# An Intron c.103-3T>C Variant of the *AMELX* Gene Causes Combined Hypomineralized and Hypoplastic Type of Amelogenesis Imperfecta: *Case Series and Review of the Literature*

**DOI:** 10.3390/genes13071272

**Published:** 2022-07-18

**Authors:** Tina Leban, Katarina Trebušak Podkrajšek, Jernej Kovač, Aleš Fidler, Alenka Pavlič

**Affiliations:** 1Department of Paediatric and Preventive Dentistry, Faculty of Medicine, University of Ljubljana, 1000 Ljubljana, Slovenia; tina.leban@mf.uni-lj.si; 2Institute of Biochemistry and Molecular Genetics, Faculty of Medicine, University of Ljubljana, 1000 Ljubljana, Slovenia; katarina.trebusakpodkrajsek@mf.uni-lj.si; 3Clinical Institute of Special Laboratory Diagnostic, University Childrens Hospital, 1000 Ljubljana, Slovenia; jernej.kovac@kclj.si; 4Department of Endodontics, Medical Faculty, University of Ljubljana, 1000 Ljubljana, Slovenia; ales.fidler@mf.uni-lj.si; 5Department of Endodontics, University Dental Clinic, University Medical Center Ljubljana, 1000 Ljubljana, Slovenia; 6Department of Paediatric and Preventive Dentistry, University Dental Clinic, University Medical Center Ljubljana, 1000 Ljubljana, Slovenia

**Keywords:** dental enamel, amelogenesis imperfecta, X-linked, exome analysis, intron variant, ultrastructure

## Abstract

Amelogenesis imperfecta (AI) is a heterogeneous group of genetic disorders of dental enamel. X-linked AI results from disease-causing variants in the *AMELX* gene. In this paper, we characterise the genetic aetiology and enamel histology of female AI patients from two unrelated families with similar clinical and radiographic findings. All three probands were carefully selected from 40 patients with AI. In probands from both families, scanning electron microscopy confirmed hypoplastic and hypomineralised enamel. A neonatal line separated prenatally and postnatally formed enamel of distinctly different mineralisation qualities. In both families, whole exome analysis revealed the intron variant NM_182680.1: c.103-3T>C, located three nucleotides before exon 4 of the *AMELX* gene. In family I, an additional variant, c.2363G>A, was found in exon 5 of the *FAM83H* gene. This report illustrates a variant in the *AMELX* gene that was not previously reported to be causative for AI as well as an additional variant in the *FAM83H* gene with probably limited clinical significance.

## 1. Introduction

Amelogenesis imperfecta (AI) covers a heterogeneous group of inherited developmental disorders that affect enamel quality and/or quantity [1]. A wide range of AI phenotypes includes signs of insufficient enamel mineralisation of varying degrees (caused by inadequate protein removal and/or insufficient mineralisation) and/or different extents of hypoplasia (ranging from, e.g., a pitted enamel surface to a profoundly reduced enamel thickness of entire tooth crowns) [2].

The estimated average global prevalence of AI is less than one in 200 [3]. However, the morbidity of this rare disease varies between countries. Epidemiological studies show that the prevalence is 43:10,000 in Turkey [4], 14:10,000 in Sweden [5], 10:10,000 in Argentina [6], 1.25:10,000 in Israel [7], and 0.7:10,000 in the United States [8]. Studies on dental abnormalities from Brazil, India, Mexico, and Turkey found 4 AI patients in a group of 478 participants [9], 3 in a group of 1123 [10], 2 in a group of 860 [11], and 1 in a group of 1200 participants [12], respectively.

AI develops due to disease-causing variants in genes involved in the process of enamel formation [13]. Amelogenesis is a complex process with several stages that are spatially and temporally precisely determined. Disturbances are reflected in the aberrant thickness, mineralisation, and/or structure of the formed dental enamel [2]. During the secretory stage, ameloblasts secrete the extracellular matrix (ECM), mainly comprised of structural proteins (e.g., amelogenin, enamelin, and ameloblastin). The ECM enables nucleations and directs the growth of hydroxyl-apatite (HA) crystals. In the secretory stage, mineralisation reaches approximately 30% by volume fractions of mature enamel in the form of thin, elongated HA ribbons. When the ECM reaches the final thickness of the enamel, amelogenesis gradually transitions into the maturation stage, in which the ECM is almost completely decomposed and removed while the HA crystals continue to thicken.

In AI patients, disease-causing variants are located in various genes related to amelogenesis: genes encoding structural proteins (*AMELX*, *ENAM*, and *AMBN*) and enamel proteases (*MMP20* and *KLK4*); proteins involved in cell adhesion (*I**TGB6*, *LAMA3*, *LAMB3*, *COL17A1*, *AMTN*, and *FAM83H*), intracellular vesicular transport (*WDR72*), and ion transport (*SLC24A4* and *STIM1*); genes that control gene expression in amelogenesis (*FAM20A* and *DLX3*); and genes encoding proteins with currently unknown functions (*GPR68*, *C4orf26*, *ACP4*, and *SP6*) [2,14]. To date, more than 20 genes are known to be involved in AI [15]. Nevertheless, despite the advancement in the diagnostics of genetic diseases, in some AI patients, the genetic aetiology remains unknown. In several studies, even after exome sequencing, disease-causing genes were confirmed only in 27% [16], 37% [17], and 49% [18] of AI patients.

Based on the phenotype and mode of inheritance, we can, to a limited extent, predict the causative gene in some AI patients. Autosomal inherited hypoplastic types of AI are associated with disease-causing variants in the following genes: *ENAM*, *LAMB3* [19], *AMBN* [20], *COL17A1* [21], *FAM20A* [22], and *ITGB6* [23]. Hypomineralised AI with an autosomal dominant mode of inheritance is commonly due to disease-causing variants in the *FAM83H* gene [2].

Hypomineralised types of AI with an autosomal recessive mode of inheritance are associated with disease-causing variants in the genes *MMP20*, *KLK4* [24], *WDR72* [25], *SLC24A4* [26], *C4orf26* [27], and *GPR8* [28]. X-linked AI results from disease-causing variants in the *AMELX* gene [29]. Depending on the position of the disease-causing variants in the *AMELX* gene, a hypomineralised or hypoplastic AI phenotype develops in patients. In female patients, stripes of normal and altered enamel are also described due to lyonisation (i.e., the inactivation of one X chromosome) [1]. The aim of this study was to evaluate the genetic characteristics of three female AI patients from two unrelated families who had a similar phenotype of intertwining clinical signs of enamel hypomineralisation and hypoplasia. The patients were carefully selected from 40 patients who were all referred due to AI between the years 2000 and 2020. In addition, we analysed the ultrastructure of the enamel of the deciduous teeth of the patients.

## 2. Materials and Methods

### 2.1. The AI Patients

Of all the AI patients referred to the University Dental Clinic in Ljubljana during the period between 2000 and 2020, only patients with similarly altered enamel quality and quantity were included in this study. At the first dental examination, each AI patient underwent a thorough dental examination, including a panoramic radiograph. Additionally, we analysed the family history of dental diseases to predict the mode of inheritance of AI in individual families. Patients were also invited to give a sample of peripheral blood for further DNA analysis and to donate an exfoliated deciduous tooth for histological observation.

The purpose of this procedure was explained in advance to each AI patient and his/her parents. All patients and, for minor patients, also their parents, voluntarily signed a consent for the collection of clinical data, blood sampling, DNA analysis, and histological analysis of the exfoliated deciduous tooth. The research protocol was approved by the Medical Ethics Committee of the Republic of Slovenia (Act No. 0120-505/2020-3).

### 2.2. Histological Analysis

For each AI patient included in the study, we collected at least one deciduous tooth brought by the patient’s parents in a saline bottle no later than three days after tooth exfoliation. Each tooth was cut in half in the bucco-lingual or bucco-palatinal direction. Samples were embedded in epoxy resin (Araldit, Ciba-Geigy, East Lansing, MI, USA) and left to polymerise overnight. The exposed longitudinal sections were first polished with raw polishing paper 500, then with fine polishing papers 800, 1200, and 4000 (Struers, Cleveland, OH, USA), and lastly with suspension with 3 μm-sized diamond grains on a wool pad (MOL Struers, Cleveland, OH, USA) and 1 μm-sized diamond grains on a napped pad (NAP Struers, Cleveland, OH, USA). Finally, the polishing was completed with an oxide polishing emulsion (OP-S Struers, Cleveland, OH, USA) on a porous neoprene pad (CHEM Struers, Cleveland, OH, USA).

The histology of the longitudinally exposed tooth was first observed with a light microscope (BXG1 PD, Olympus, Japan). Samples were washed with distilled water for 30 s, dried with compressed air, dehydrated with 96% alcohol, dried again, and sputter-coated in a vacuum with a thin carbon layer (Bal-Tec SCD 050 Sputter Coater, Scotia, NY, USA). On re-examination, some samples were polished again and etched with 37% orthophosphoric acid for 60 s. The enamel microstructure was observed using scanning electron microscopy (SEM) (JEOL JSM-5610, JEOL, Tokyo, Japan) under 10 kV, with secondary electron imaging (SEI).

### 2.3. Molecular Genetic Analysis

For each patient, DNA was isolated from 3 mL of peripheral blood with a FlexiGene DNA kit (Qiagen, Hilden, Germany). Whole exome sequencing (WES) was performed using a standardised series of procedures, starting with the capture of exome sequences using the capture kit Agilent SureSelect Human All Exon v5 (manufactured by Agilent Technologies, Santa Clara, CA, USA) (http://cshprotocols.cshlp.org/; accessed on 14 September 2021), following the manufacturer’s instructions. This was followed by whole-exome sequencing (WES) on a NovaSeq 6000 platform (Illumina), obtaining about 250 nucleotide long paired-end readings. Data were analysed with the analytical software bcbio-nextgen—v.1.2.8. (https://doi.org/10.5281/zenodo.3564938, accessed on 14 September 2021). Reads were aligned to the NCBI reference genome using the Burrow–Wheeler aligner (http://bio-bwa.sourceforge.net/, accessed on 14 September 2021). For data visualisation, we used the Integrative Genomics Viewer (Broad Institute, University of California) [30]. Based on segregation and the use of computer programs (SIFT, Polyphen2, CADD, Human Splicing Finder), we predicted the effect of the sequence variant found within genes associated with AI. The pathogenicity of the variants was evaluated according to the recommendation of the American College of Medical Genetics (ACMG) [31]. Regions identified as causative by WES were further confirmed/analysed with polymerase chain reaction (PCR) and Sanger sequencing. Such areas were found near exon 4 of the *AMELX* gene and in exon 5 of the *FAM83H* gene. For PCR amplifications, we designed oligonucleotide primers based on the reference sequence of the *AMELX* gene (NG_012040.1) and the *FAM83H* gene (NG_016652.1) using the Primer-BLAST software (http://www.ncbi.nlm.nih.gov/tools/primerLblast/, accessed on 9 February 2022). The primer sequencing and cycling condition are described in Appendix A. PCR amplicons were sequenced using the ABI 3500 Genetic Analyser (PE Applied Biosystems, Piscataway, NJ, USA). Results were compared with the BlastN protocol (accessed on 22 February 2022) to normal sequences of the *AMELX* (NG_012040.1) and *FAM83H* (NG_016652.1) genes available online (http://www.ncbi.nlm.nih.gov).

## 3. Results

### 3.1. Clinical and Radiographic Findings

Among the referred patients with AI, three patients from two unrelated families, both Caucasian, met the inclusion criteria (Table 1; no. 39/F27 represents a patient from family I, and the patients listed under numbers 36/F25 and 37/F25 are from family II). Clinically, all three probands (an almost 10-year-old girl from family I as well as an 11-and-a-half-year-old girl and her not-yet 7-year-old sister from family II) from two families had similar aberrant tooth crowns featuring hypoplastic defects and markedly hypomineralised enamel that chipped away on surfaces exposed to occlusal forces. In family I and in family II, the proband’s mother (II.4) and the probands’ father (III.6), respectively, had similarly affected teeth (Appendix A). The teeth of the other family members in both families were reported to be normal.

All deciduous and permanent teeth of the 9-year-and-10-month-old girl from family I showed rough and uneven surfaces of hypomineralised enamel (Figure 1A–C). The hypoplastic enamel had a yellowish to brownish discolouration. Uneven surfaces promoted dental plaque retention; marginal gingivitis was present. The probing had also delayed eruption. Radiographically, there was a poor contrast between enamel and dentine (Figure 1D). The pedigree in family I segregated with an X-linked mode of inheritance (Figure 1E). The girl’s grandfather (I.2), mother (II.4), and three maternal sisters (II.7, II.9, and II.11) but none of the maternal brothers were reported to have developmentally altered enamel.

In family II, a similarly aberrant clinical phenotype of the enamel of all teeth was observed in an 11-year-and-6-month-old girl (Figure 2A–C) and her 6-year-and-9-month-old sister (Figure 2E–G). In both sisters, the enamel of all permanent and deciduous teeth showed poorly mineralised and hypoplastic enamel. In some areas, chalky-white-coloured teeth turned yellowish-brown. On the occlusal surfaces of the upper and lower deciduous molars of the younger sister, evidence of a widespread post-eruptive enamel loss was detected; enamel tended to chip away due to attrition (Figure 2F,G). On the radiographic image, less mineralised enamel was reflected in the reduced contrast between the enamel and dentin (Figure 2D,H). Pedigree analysis suggested an X-linked inheritance pattern (Figure 2I).

### 3.2. Histological and Ultrastructural Analysis

Light microscopy analysis of a lower left deciduous first molar (tooth 74) of the girl from family I confirmed hypoplastic and less-mineralised enamel (Figure 3A). Under scanning electron microscopy (SEM), pits and an unusual crater-like appearance were visible on the buccal surface of the enamel (Figure 3B). A higher degree of porosity was expressed through the whole enamel thickness, where individual sponge-like structures were recognisable (Figure 3C). Similarly, the enamel ultrastructure of an upper left deciduous second molar (tooth 65) of the older girl from family II was altered: the uneven surface of the enamel showed pronounced pits (marked with a yellow arrow in Figure 3D) and insufficient mineralisation throughout the entire thickness of the enamel. A cross-section of the enamel showed the inclusion of unusual globular voids (Figure 3E,F). In both tooth samples, two layers of enamel were evident (marked with asterisks in Figure 3A,D), built before and after birth. The inner layer revealed a more organised prism structure and seemed better mineralised compared to the outer layer.

### 3.3. Molecular Genetic Analysis

Exome sequencing in family I and family II identified the intron variant NM_182680.1: c.103-3T>C of the *AMELX* gene. This variant, located only three nucleotides before exon 4, has not been previously reported in AI patients. It is listed in the dbSNP database (rs1271593349; https://www.ncbi.nlm.nih.gov/snp/, accessed on 22 February 2022) and in the VarSome database (VarSome The Human Genomics Community; https://varsome.com, accessed on 22 February 2022) as a variant with uncertain significance (VUS) but is not listed in the gnomAD browser (https://gnomad.broadinstitute.org, accessed on 22 February 2022). We attempted to predict the impact of this variant on splicing. Nevertheless, the majority of in silico prediction tools, i.e., Human Splicing Finder (https://www.umd.be/HSF/, accessed on 22 February 2022) [32], Alternative Splice Site Predictor (https://wangcomputing.com/assp/, accessed on 22 February 2022) [33], and NetGene2-2.42 (https://services.healthtech.dtu.dk/service.php?NetGene2-2.42, accessed on 22 February 2022) [34], recognised neither the wild-type sequence nor the altered sequence as a possible acceptor splice site. The exception was Ex-Skip (https://ex.skip-img.cas.cz, accessed on 22 February 2022) [35], which predicted a higher probability of exon skipping mutations compared to the wild-type sequence. In addition, the sequence is perfectly conserved among a wide range of vertebrate orthologs (9 primates/43 eutherian mammals), which are closer on the evolutionary scale (https://ensembl.org; accessed on 22 February 2022). According to the ACMG recommendation, the variant was classified as a VUS with the following grade: PM2 (moderate), extremely low frequency in gnomAD population databases. Sanger sequencing confirmed that the proband in family I (III.6) was heterozygous for this variant. Furthermore, it showed the co-segregation of the variant with the clinical characteristics in her mother (II.4). In family II, this variant was confirmed in the index patient (IV.4) and in her younger sister (IV.5) in heterozygous form and in their father (III.6) in hemizygous form. This confirmed the co-segregation of this variant with the clinical presentations of all analysed affected members of families I and II (Appendix A).

In addition, the proband in family I (III.6) carried an additional heterozygous variant: NM_198488.5: c.2363G>A in the *FAM83H* gene. This was a missense variant located in exon 5 of the *FAM83H* gene that changes guanine to adenine. It results in the replacement of a polar uncharged serine by an equally polar uncharged asparagine at amino acid position 788 (p.Ser788Asn). Polyphen-2 predicted it to be “probably damaging” with a score of 0.991 (sensitivity: 1.00; specificity: 0.00) [36] and Sift predicted it to affect protein function (deleterious) with a score of 0.01 [37]. According to the ACMG recommendation, the variant was classified as benign with the following grades: BA1 (standalone), GnomAD allele frequency greater than the 0.05 threshold; BP6 (supporting), ClinVar classifies this variant (rs56148058) as benign. Confirmatory Sanger sequencing confirmed that the girl (III.6) was heterozygous for this variant. Furthermore, it showed the co-segregation of the variant with the disease in her mother (II.4) (Appendix A).

### 3.4. Literature Review

The reported disease-causing *FAM83H* gene variants and related clinical characteristics are reviewed in Table 2. In Figure 4, the location of the amino acid affected in the individual variant is described according to the protein domain. The reported disease-causing *AMELX* gene variants and related clinical characteristics are reviewed in Table 3 and presented in Figure 5.

**Table 2 genes-13-01272-t002:** Review of the disease-causing *FAM83H* gene (NM_198488.5) variants and related clinical characteristics.

c DNA	Protein	Enamel Phenotype	Reference
c.860C>A	p.S287X	Generalised hypocalcified	[38]
c.891T>A	p.Y297X	Generalised hypocalcified	[39]
c.906T>G	p.Y302X	Hypocalcified; hypocalcified, polished-looking enamel surface	[40]; [41] *
c.923dupT	p.V309RfsX16	Hypocalcified	[41]
c.923_924delTC	p.L308RfsX16	Generalised hypocalcified	[38]
c.931dupC	p.V311RfsX14	Hypocalcified	[42]
c.973C>T	p.R325X	Hypocalcified	[41,42,43,44]
c.1024T>A	p.S342T	Hypocalcified	[45]
c.1130_1131delinsAA	p.S377X	Hypocalcified	[42]
c.1147G>T	p.E383X	Hypocalcified	[42]
c.1192C>T	p.Q398X	Hypocalcified; not described *	[43,46,47]; [17] *
c.1222A>T	p.K408X	Hypocalcified	[48]
c.1243G>T	p.E415X	Generalised hypocalcified	[39]
c.1261G>T	p.E421X	Generalised hypocalcified	[49]
c.1282C>T	p.Q428X	Hypomineralised	[16]
c.1289C>A	p.S430X	Not described *; hypomature	[17] *; [16]
c.1330C>T	p.Q444X	Hypocalcified	[46,47]
c.1354C>T	p.Q452X	Generalised hypocalcified	[18,40,41,50,51]
c.1366C>T	p.Q456X	Hypocalcified	[47]
c.1369C>T	p.Q457X	Hypocalcified	[51]
c.1374C>A	p.Y458X	Generalised hypocalcified, small focal areas of normal-looking enamel	[52]
c.1379G>A	p.W460X	Generalised hypocalcified	[38,53]
c.1380G>A	p.W460X	Generalised hypocalcified	[39]
c.1387C>T	p.Q463X	Hypocalcified	[54]
c.1408C>T	p.Q470X	Generalised hypocalcified	[38]
c.1669G>T	p.G557C	Hypocalcified, attenuated	[55]
c.1872_1873delCC	p.L625AfsX79	Localised hypocalcified	[38]
c.1915A>T	p.K639X	Hypocalcified	[51]
c.1993C>T	p.Q665X	Generalised hypoplastic/hypomineralisation; less severe phenotype	[56]
c.2029C>T	p.Q677X	Generalised hypocalcified; not described *	[18,39,41,56]; [16,17] *
c.2080G>T	p.E694X	Localised hypocalcified	[38]

* not available/not clearly defined by the authors.

**Figure 4 genes-13-01272-f004:**
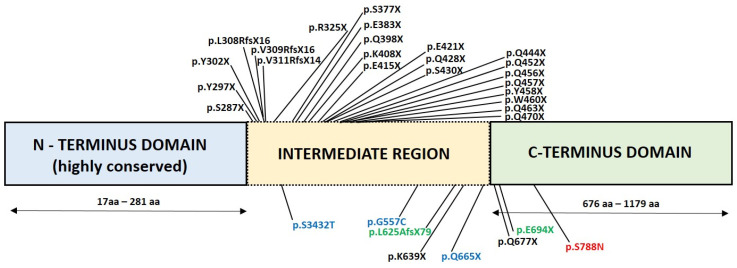
Diagram showing disease-causing variants, named according to the reference sequence NM_198488.5, identified in reported families with hypocalcified AI. The *FAM83H* domains are shown in coloured boxes. The bottom line shows the range of amino acids in particular domains (blue: highly conserved N-terminus domain encompassing amino acids 17–281; yellow: intermediate region encompassing amino acids 282–675; green: highly conserved C-terminus domain encompassing amino acids 676–1179). The disease-causing variants shown in black are associated with the generalised phenotype, those in green with the localised phenotype, and those in blue with the attenuated phenotype. The variant identified in this study is shown in red.

**Table 3 genes-13-01272-t003:** Review of the disease-causing variants described in the *AMELX* gene (NM_182680.1) and related clinical characteristics.

cDNA	Protein	Enamel Phenotypes	Reference
c.-11311600_X705268del	p.0?	Hypoplasia/hypomineralisation	[57]
c.-39356_X6166del	p.0?	Snow-capped appearance	[58]
c.-21552_X67556del	p.0?	Snow-capped appearance	[58]
c.2T>C	p.M1T	Hypoplasia	[59]
c.11G>A	p.W4X	Hypoplasia	[60]
c.11G>C	p.W4S	Hypoplasia	[59]
c.13_22delATTTTATTTG	p.I5PfsX41	Hypoplasia	[61]
c.103-3T>C		Hypoplasia/hypomineralisation	*In this study*
c.120T>C	p.A40A	Hypoplasia/hypomineralisation	[62]
c.143T>C	p.L48S	Hypoplasia/hypomineralisation	[15]
c.152C>T	p.T51I	Hypoplasia/hypomineralisation	[63]
c.155C>G	p.P52R	Hypoplasia	[64]
c.155C>T	p.P52L	Hypoplasia	[16]
c.155delC	p.P52LfsX2	Hypoplasia/hypomineralisation	[17,65,66]
c.185delC	p.P62RfsX47	Hypoplasia/hypomineralisation	[67]
c.208C>A	p.P70T	Hypomaturation	[17,18,68,69,70]
c.230A>T	p.H77L	Hypomaturation	[1]
c.242C>T	p.P81L	Hypoplasia	[29]
c.385delC	p.H129TfsX60	Hypoplasia	[71]
c.420delC	p.Y141TfsX48	Hypoplasia	[72]
c.473delC	p.P158HfsX31	Hypoplasia; **	[63]; [17] **
c.518delC	p.P173LfsX16	Hypoplasia	[56]
c.541delC	p.L181CfsX8	Hypoplasia	[1,73]
c.571G>T	p.E191X	Hypoplasia	[63]
c.X362843_X367565del	p.?	Hypomineralisation	[74]

** not available/not clearly defined by the authors.

**Figure 5 genes-13-01272-f005:**
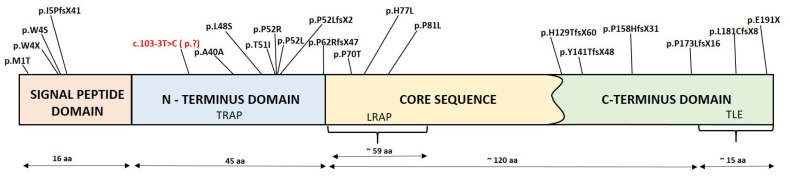
Diagram showing disease-causing variants, named according to the reference sequence NM_182680.1, identified in reported families with the *AMELX* phenotype. The amelogenin protein has several structural domains shown in coloured boxes (adapted from [75,76]). The bottom line shows the range of amino acids in particular domains (TRAP represents tyrosine-rich amelogenin protein, LRAP leucine-rich amelogenin protein, and TLE telopeptide). The disease-causing variant identified in this study is shown in red. * Disease-causing variants with no product are not labelled ([57,58,74]).

## 4. Discussion

This study aimed to characterise the genotypes and histology of the enamel of female AI patients with similar clinical and radiographic findings from two unrelated families. Since all affected individuals showed a similar phenotype, we expected to confirm similar enamel histology and the same genotype. While the results of the histological analysis were similar, the results of the genetic analysis showed some differences. Genetic analysis confirmed the same disease-causing variant in the *AMELX* gene in all AI-affected individuals from both families; this variant was not previously reported to be causative for AI. An additional variant in the *FAM83H* gene was detected in one of the families, although it probably has limited clinical importance.

To date, *AMELX* (OMIM *300391) is the only known gene that causes X-linked AI, with more than 20 disease-causing variants reported thus far, as reviewed in Figure 2 and Table 2. Based on the predicted consequences of *AMELX* alterations, which cause different variants of amelogenin proteins and consequently different clinical pictures, Wright and colleagues reported a genotype–phenotype correlation (2003). The disease-causing variants with localisation in the signal peptide domain cause a total loss of the amelogenin protein and are expressed as hypoplastic AI. When present in the N-terminus domain and the C-terminus domain, they cause hypomineralised-hypomaturated and hypoplastic AI, respectively. Furthermore, in females with *AMELX* disease-causing variants, patches or ribbons of normal and altered enamel, i.e., lyonisation, may be observed [77]. In male patients, the enamel is more severely affected than in females with the same *AMELX* disease-causing variants. The transcriptions and translations of the *AMELY* gene represent only one-tenth those of the *AMELX* gene, resulting in the extensive deficiency of amelogenin proteins in men with the AMELX variant [75].

Amelogenins are the most abundant structural proteins of developing enamel. During amelogenesis, the *AMELX* gene is translated into several transcripts through the process of alternative splicing, which results in several protein isoforms. In humans, there are six isoforms of the *AMELX* gene [29]. The most abundant mRNA is an exon 4-skipped full-length transcript, which has been extensively studied in mouse models [78]. Generating a transgenic animal model showed that the inclusion of exon 4 of the *AMELX* gene, caused by a specific disease-causing variant, results in hypomineralised enamel with pitted hypoplastic regions [62]. This could imply that transcripts that either include or exclude exon 4 have different functional roles. It seems that for normal enamel formation, it is essential that only a small proportion of amelogenin proteins include the exon 4 transcription. Indeed, a silent variant (NM_182680.1: c.120T>C; p.A40A) [62] and a missense variant (NM_182680.1: c.143T>C; p.L48S) [15] of the *AMELX* gene are both reported to alter the RNA splicing and inclusion of exon 4, resulting in hypomineralised AI with the pitted hypoplastic phenotype.

The AI phenotype observed in both families reported here was a combination of hypoplastic and hypomineralised. The *AMELX* variant NM_182680.1: c.103-3T>C, identified in the affected members of both families, is located in the canonical splice acceptor site of exon 4. In general, variants that alter the nucleotides within canonical splice sites may interfere with the accurate splicing of an intron, leading to exon skipping, intron retention, or, in some cases, the utilisation of a cryptic splice site [79]. As a result, transcription and translation into functional proteins are altered or even disabled. While the intronic variant reported here is absent from the general population and is conserved among the vertebrates, only one in silico tool predicted it to be pathogenetic, while others could not identify the normal or altered sequence as potentially important in splicing. Nevertheless, the variant co-segregated with the disease in all analysed patients from both families, which strongly indicates the pathogenicity of this variant. Furthermore, we aimed to definitively confirm its pathogenicity with mRNA analysis. Nevertheless, due to extremely low AMELX mRNA content in the peripheral blood of the participating AI patients and in the pulp tissue of their deciduous teeth, we did not isolate enough AMELX mRNA to confirm or refute earlier claims.

The variant c.2363G>A of the *FAM83H* gene (OMIM *611927), which was found along with the intron variant c.103-3T>C of the *AMELX* gene in both probands from family I, was located in the fifth exon, like all the disease-causing variants found thus far in the *FAM83H* gene. Most previously reported variants are nonsense or frameshift disease-causing variants; introducing a premature stop codon between the serine at position 287 and the glutamate at position 694 results in a truncated protein [47]. In the variant reported here, one amino acid had been replaced (p.S788N), most likely resulting in a more attenuated and masked phenotype. Nevertheless, this particular variant is predicted to be benign due to its high presence in the general population. In AI patients from family I, in whom we confirmed the combined heterozygous variants *AMELX* c.103-3T>C and *FAM83H* c.2363G>A, the phenotype was decisively influenced by a variant of the *AMELX* gene. In this study, all AI patients from both families had similar enamel phenotypes. We argue that having the same *AMELX* variant and an X-linked mode of inheritance was the cause of AI. Nevertheless, predicting the combined effect of such dual genetic variants on clinical presentation is extremely difficult.

Accordingly, the histological findings from the enamel of the exfoliated deciduous teeth of the AI-affected female from family I with intron variant c.103-3T>C of the *AMELX* gene and variant c.2363G>A of the *FAM83H* gene were similar to those of the AI-affected female from family II with intron variant c.103-3T>C of the *AMELX* gene. In both samples, the enamel surface was rough, with various pits included. A cross-section of the enamel showed the porous bulk of the enamel with the inclusion of individual globular voids, especially in the outer layer of enamel. In the literature, histological findings from the enamel of X-linked AI patients are limited. In AI-affected patients with variants located in the signal peptide domain, histology displays profoundly hypoplastic AI, i.e., thin enamel that lacks prismatic structure [59], while in those with an abnormality located in the N-terminus domain, the enamel surface is rough and chalky with exposed enamel prisms at the bottom of the pits [67]. In both genders, the enamel shows similarly altered histology; in males, however, aberrations are more pronounced [70], while in females, vertical bands of rather unaffected enamel prisms and less densely packed enamel prisms are described [67]. The enamel histology of patients with *AMELX* variants in the C-terminus domain has not been reported so far.

Reports on enamel histology in AI patients with *FAM83H* variations also differ. Variants translated in a protein with less than 694 amino acids result in defective enamel prisms, especially those at the dentine–enamel junction (DEJ), and in increased organic content within the formed enamel [44,52]. Variants resulting in at least 694 amino acid proteins result in an almost regular prismatic structure with areas of amorphous material, possibly proteins [38]. This suggests that a truncated protein of approximately over 694 amino acids conserves some residual activity during amelogenesis.

In the probands reported here, tooth samples showed porous enamel with pitted surfaces. Interestingly, in both tooth samples, a neonatal line separated prenatally and postnatally formed enamel of distinctly different mineralisation qualities. A similar observation was registered in patients with the disease-causing variant c.120T>C of the *AMELX* gene [62]. They reported an atypical structure with crater-shaped voids and irregular prism organisation in the bulk of the enamel; only near the DEJ was normal-looking, well-organised enamel found. Normally, full-length amelogenins assemble into spheres forming a protein matrix scaffold, essential in the proper elongation and formation of enamel crystals [80]. Due to the alternative splicing of the primary mRNA, more types of amelogenin isoforms can be expressed and secreted in the ECM [76]. In mouse models, the majority of amelogenin isoform spheres translated without exon 4 are sufficient to achieve normal enamel thickness [78]. Amelogenin isoforms with exon 4 included are believed not to be required for the initial enamel mineralisation [62]. Moreover, the results of transgenic mouse models show that amelogenin isoforms with exon 4 included are secreted at the specific time of amelogenesis only, i.e., during the early maturation stage [81]. It seems that individual amelogenin isoforms have different functions and are produced at precisely determined times and/or locations during amelogenesis; it is possible that the abundant translation of amelogenin isoforms with included exon 4 manifests in impaired enamel mineralisation with pitted enamel surfaces.

In conclusion, we report the intronic variant c.103-3T>C of the *AMELX* gene associated with X-linked AI in three patients from two families. All subjects had similar hypomineralised AI phenotypes with hypoplastic regions and comparable enamel histology findings. The clinical significance of the c.2363G>A variant in the *FAM83H* gene remains to be revealed.

## Figures and Tables

**Figure 1 genes-13-01272-f001:**
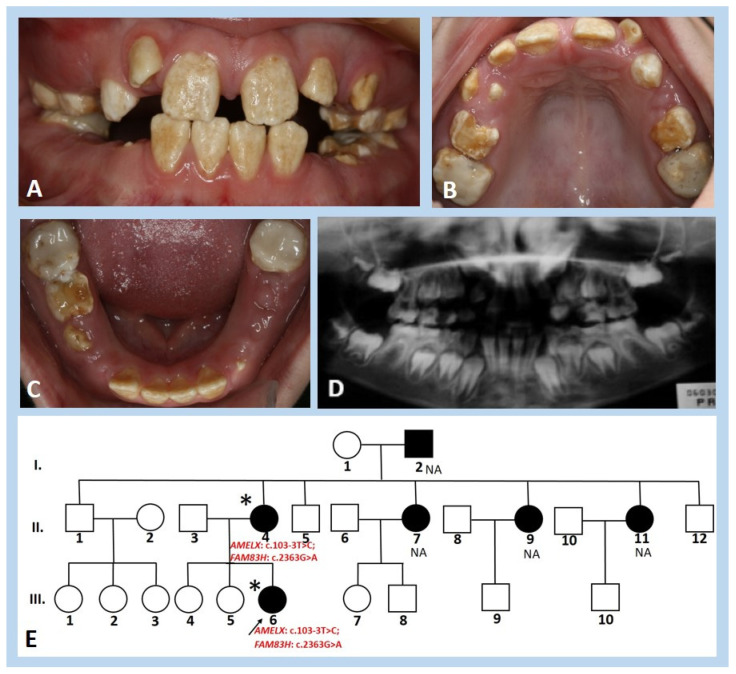
Clinical photographs, panoramic radiograph, and pedigree of the girl from family I. (**A**–**C**) Mixed dentition of the 9-year-and-10-month-old probed (III.6) reveal the chalky-white to yellowish hyperplastic enamel of all deciduous and permanent teeth. On the occlusal surfaces of upper and lower deciduous molars and permanent first molars, profound attrition is observed. Permanent first molars are covered with extensive glass ionomer fillings. All first deciduous molars and the lower right second deciduous molar were extracted. Both upper lateral permanent incisors are erupting ectopically. (**D**) Panoramic radiograph shows the presence of all permanent germs (wisdom teeth included). Those teeth that have developing tooth crowns not erupted exhibited normal anatomy, with enamel of normal thickness but lacking contrast between enamel and dentine. (**E**) Pedigree of family I shows the presence of an X-linked mode of inheritance. The arrow indicates the girl described (III.6); the symbol (*) indicates participating individuals; NA indicates not-available family members.

**Figure 2 genes-13-01272-f002:**
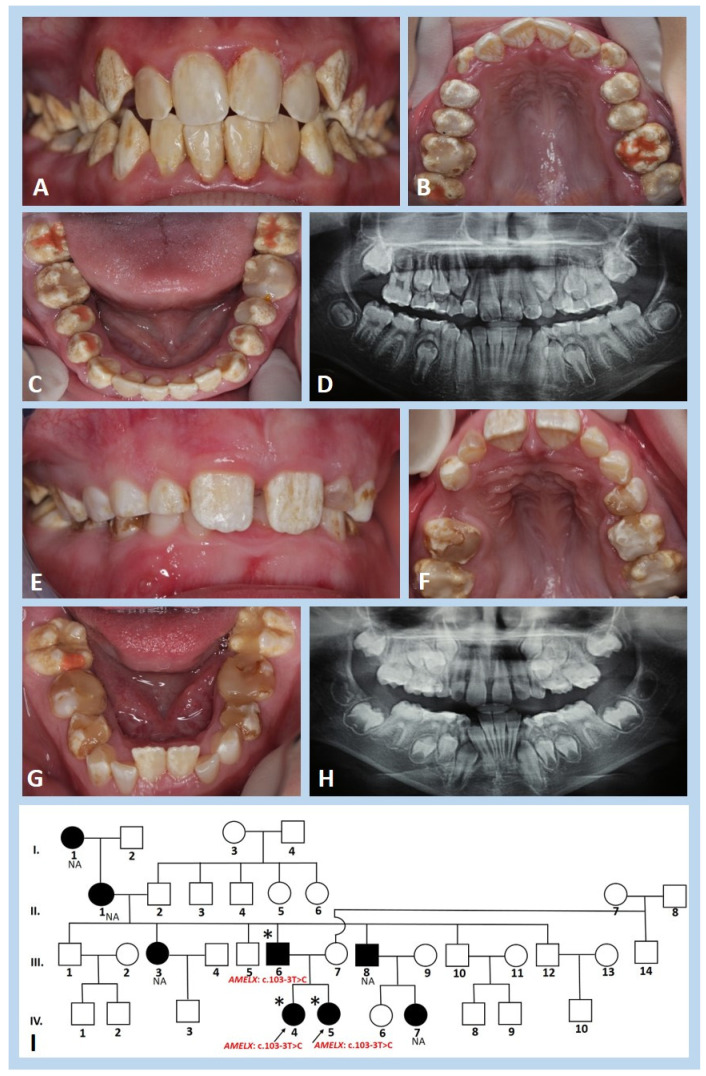
Clinical photographs, panoramic radiographs, and pedigree of the girls from family II. (**A**–**C**) Hypomineralised mixed dentition of the 11-year-and-6-month-old girl (IV.4) also exhibits some hypoplastic areas on all tooth crowns. Teeth are whitish with diffuse chalky-like patches, in some areas altering to yellow-brownish. All permanent incisors are restored with composite resins and all premolars and molars with glass ionomers. (**E**–**G**) The enamel of the 6-year-and-9-month-old sister (IV.5) is similarly altered, of chalky to yellowish colour. All deciduous and permanent molars are covered with extensive temporary fillings. (**D**,**H**) The panoramic radiographs show the enamel of adequate thickness, but with radiopacity, similar to dentine. (**I**) The filled symbols denote individuals of four generations of family II who have similarly affected enamel. Arrows indicate both girls examined (IV.4 and IV.5), the symbol (*) indicates family members who participated in the study, and NA indicates non-available family members.

**Figure 3 genes-13-01272-f003:**
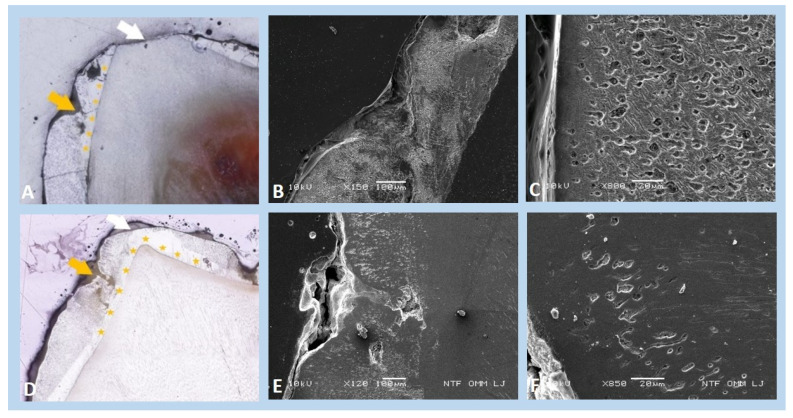
Images of (**A**–**C**) an exfoliated and etched lower deciduous first molar (tooth 74) of the girl (III.6) from family I and (**D**–**F**) a non-etched upper deciduous second molar (tooth 65) of the older girl (IV.4) from family II. (**A**,**D**) Light microscopy reveals a pitted enamel surface (yellow arrows) that is also missing in some areas due to attrition (white arrows). Two layers of enamel are visible, with the inner layer (asterisks) being better mineralised (etched, ×60 and non-etched, ×100, respectively). (**B**,**E**) Scanning electron microscopic (SEM) images of these same surfaces of both affected teeth show rather normal structures in the inner enamel layer and irregular histology with numerous voids and amorphous artefacts in the outer layer (etched, ×150, SEI and non-etched, ×120, SEI, respectively). (**C**,**F**) Under higher magnification, we see profoundly porous enamel and a reduced number of enamel prisms with increased inter-prism space in the outer layer (etched, ×800, SEI and non-etched, ×850, SEI).

**Table 1 genes-13-01272-t001:** Clinical characteristics of all referred AI patients in the period between 2000 and 2020. Patients included in further analysis are highlighted in blue.

Number	Family (F)	Gender	Predicted Mode of Inheritance	Phenotype *	
				Hypomineralisation Type	Hypoplastic Type
1	F1	Female	**		+
2	F2	Female	**		+
3	F3	Male	AD	+	
4		Male		+	
5	F4	Male	AD		+
6		Female			+
7		Female			+
8		Male			+
9	F5	Female	**		+
10	F6	Female	**		+
11	F7	Female	AD		+
12		Male			+
13	F8	Male	AD	+	
14		Female		+	
15	F9	Female	AD		+
16	F10	Female	**		+
17	F11	Female	AR	+	+
18	F12	Male	AR	+	
19	F13	Male	AD	+	+
20		Female		+	+
21	F14	Female	AR	+	
22	F15	Male	AD	+	+
23	F16	Male	AD	+	+
24		Male		+	+
25	F17	Female	AD	+	+
26		Male		+	+
27	F18	Female	AR	+	
28	F19	Female	**		+
29	F20	Male	AD		+
30		Male			+
31	F21	Male	AD		+
32	F22	Female	**		+
33	F23	Female	**		+
34	F24	Male	AD	+	
35		Male		+	
**36**	**F25**	**Female**	**X-linked**	**+**	**+**
**37**		**Female**		**+**	**+**
38	F26	Male	**	+	
**39**	**F27**	**Female**	**X-linked**	**+**	**+**
40	F28	Female	**		+

* According to observed prevalent clinical signs; ** Not possible to determine due to the small number of family members. (+) presence of enamel hypoplastic and/or hypomineralised clinical signs; Patients numbered 36, 37, and 39 (lines in bold) were included in this study because their dental examination confirmed hypoplastic and hypomineralised enamel.

## Data Availability

The data presented in this study are available on request from the corresponding author. The data are not publicly available due to privacy restrictions.

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
