# Peer review of "An Intron c.103-3T>C Variant of the AMELX Gene Causes Combined Hypomineralized and Hypoplastic Type of Amelogenesis Imperfecta: Case Series and Review of the Literature"

_genes, 2022, doi:10.3390/genes13071272_

Round 1

Reviewer 1 Report

 I appreciate the opportunity to review an article entitled “An intron c.103-3T>C variant of the AMELX Gene Causes Combined Hypomineralized and Hypoplastic Type of Amelogenesis Imperfecta: Case series and Review of the Literature”. Amelogenesis Imperfecta is a rare disease. We are falling behind from having the whole picture of the disease. The accumulation of each case, therefore, will be important to grasp this rare desease. I put a point I have noticed below at this time. I look forward to read your next paper using mice having c.103-3T>C mutation.

1)The names of many AI-related genes are given as the abbreviations in     “Intriduction”.   Their full names should be listed.  

Author Response

We are thankful to the reviewer for the very positive assessment of our work and the manuscript, and for the comment. Consistent with this commentary, we have provided the full name below the Abbreviations for each of these abbreviations of the listed AI-related genes.

Reviewer 2 Report

This manuscript reported two new cases of X-linked amelogenesis imperfecta. The authors provided detailed clinical and genetic data to demonstrate probands phenotype and new SNPs and summarized previously reported SNPs of AMELX and FAM83H by literature review. In these decades, novel SNPs are found followed by technical improvement. This helps subcategorizing clinical features which may support clinicians to diagnose AI. However, there are couple of points which are needed to improve.

1.     Line 74-107, table 1 and 2, Figure 1 and 2: These sentence, tables, and figures are results of the literature review but not introduction. Thus, they should be moved to after line 290 under a new subtitle “Literature review”.

2.     Line 71-: there is no description about hypomaturation AI (type II) and hypomature and hypoplastic enamel with taurodontism (type IV). Please describe more detail information of each AI type.

3.     Materials and Methods. The AI patients: Please summarize the detailed patient’s information; how many patients/families were recruited, gender, age, and race/genetic background.

4.     Table 3: Which families (#1-28) corresponds to family I and II? Which patients (#1-40) corresponds to the probands in Figure 3 and 4?

5.     Figure 5: Figure 5E and F are not typical acid-etched enamel structure images. They seem non-etched SEM. There are no backscattered SEM images elsewhere although the authors described in “Materials and Methods”.

6.     Figure S2: The control (not affected) sanger sequencing data are lacked in Family I.

7.     Discussion: Please discuss how much and what additional phenotypic impacts from FAM83H mutation in the probands of family I compared with ones in family II?

Author Response

Comment 1:  Line 74-107, table 1 and 2, Figure 1 and 2: These sentence, tables, and figures are results of the literature review but not introduction. Thus, they should be moved to after line 290 under a new subtitle “Literature review”.

We have changed the order of the text in accordance with this comment. Tables and figures are also renumbered accordingly.

Comment 2: Line 71-: there is no description about hypomaturation AI (type II) and hypomature and hypoplastic enamel with taurodontism (type IV). Please describe more detail information of each AI type.

A series of classifications make an attempt to classify a genetically and phenotypically heterogeneous group of AI diseases, but neither is ideal. One of the frequently used classification (Wikop, 1989) divide AI diseases accordingly to phenotype, x-ray findings (taurodontism) and mode of inheritance into 4 groups: type I-hypoplastic, type II-hypomature, type III-hypocalcified and type IV-hypomature-hypoplastic with taurodontism (which are further divided into subgroups). In Witkop classification type II and type III, are both classified as hypomineralized AI.

That many different classifications of AI are proposed can also be seen in Tables 1 (now Table 2) and Table 2 (now Table3), where we listed the originally published data. Regarding the classification of AI patients, we followed the suggestion of Aldred and Crawford (1995). In this perspective descriptions of types II and IV would make no sense.

Comment 3: Materials and Methods. The AI patients: Please summarize the detailed patient’s information; how many patients/families were recruited, gender, age, and race/genetic background.

Accordingly to this comment, the following words have been added – the first paragraph of 3.1. Clinical and radiographic findings (added words are bold and underlined):

»Among the referred patients with AI, three patients from two unrelated families, both Caucasian, met the inclusion criteria (Table 1; no. 39/F27 represents a patient from family I, and patients listed under numbers 36/F25 and 37/F25 are from family II). Clinically, all three probands (an almost 10-year-old girl from family I, an 11-and-a-half-year-old girl and her not-yet 7-year-old sister from family II) from two families had similar aberrant tooth crowns: hypoplastic defects and markedly hypomineralized enamel that chip-away on surfaces exposed to occlusal forces…. «

Comment 4: Table 3: Which families (#1-28) corresponds to family I and II? Which patients (#1-40) corresponds to the probands in Figure 3 and 4?

Please see the previous answer: the reply to this comment is given in the prior answer.

Comment 5: Figure 5: Figure 5E and F are not typical acid-etched enamel structure images. They seem non-etched SEM. There are no backscattered SEM images elsewhere although the authors described in “Materials and Methods”.

We thank the reviewer for pointing out the missing mark D (underlined) in Figure 5 – now Figure 3. Indeed, previously numbered Figures 5D, 5E and 5F are not acid-etched. Please note that this is stated in the description (marked with yellow):

»Figure 3: Images of (A-C) an exfoliated and etched lower deciduous first molar (tooth 74) of the girl (III.6) from family I and (D-F) a non-etched upper deciduous second molar (tooth 65) of the girl (IV.4) from family II. (A, D) Light microscopy reveals a pitted enamel surface (yellow arrows) that is also missing in some areas due to attrition (white arrows). Two layers of enamel are visible with the inner layer (asterisks) being better mineralized (etched, x60 and non-etched, x100, respectively). (B, E) Scanning electron microscopic (SEM) images of these same surfaces of both affected teeth show rather normal structure in the inner enamel layer and irregular histology with numerous voids and amorphous artefacts in the outer layer (etched, x150, SEI and non-etched, x120, SEI, respectively). (C, F) Under higher magnification, profoundly porous enamel and a reduced number of enamel prisms with increased inter-prism space are evident in the outer layer (etched, x800, SEI and non-etched, x850, SEI).«

Thank you also for the comment regarding backscattered SEM images. Although backscattered SEM images were also taken, none of them is included in the manuscript. Accordingly, the term is omitted from the text as follows – the last sentence of the second paragraph of 2.2. Histological analysis (marked bold and crossed out):

»…. Enamel microstructure was observed using scanning electron microscopy (SEM) (JEOL JSM-5610, JEOL, Tokyo, Japan) under 10 kV, with secondary electron imaging (SEI) and the backscattered electron imaging (BSE)

Comment 6: Figure S2: The control (not affected) sanger sequencing data are lacked in Family I.

In family I, only the proband and the proband’s mother were available for a dental examination and genetic analysis. As explained under Figure S1, a dental examination of other siblings (both adults) and the father from family I was not possible as they moved away.

Comment 7: Discussion: Please discuss how much and what additional phenotypic impacts from FAM83H mutation in the probands of family I compared with ones in family II?

The commentary pointed-out an interesting aspect of this paper, which was also addressed by us, the authors of the paper. In the discussion, we pointed out that the phenotypic impact of the FAM83H mutation in the family I probands compared to those in family II is "probable of limited clinical importance" (last sentence of the first paragraph). Further on, we discussed this issue in the parts of the sixth and seventh paragraphs.

Comment: Moderate English changes required

A native English-speaking edited the manuscript. Accordingly, we have added an acknowledgement as follows:

“Acknowledgments: The authors would like to sincerely thank Ms. Nika Breskvar, Ms. Jurka Ferran and Mr. Samo Smolej for their expert technical assistance. In addition, we thank Mr. Terry Troy Jackson for expert editing of the manuscript.